# Experimental Treatments for Spinal Cord Injury: A Systematic Review and Meta-Analysis

**DOI:** 10.3390/cells11213409

**Published:** 2022-10-28

**Authors:** Farihah Iqbal Khan, Zubair Ahmed

**Affiliations:** 1Institute of Inflammation and Ageing, University of Birmingham, Edgbaston, Birmingham B15 2TT, UK; 2Centre for Trauma Sciences Research, University of Birmingham, Edgbaston, Birmingham B15 2TT, UK

**Keywords:** spinal cord injury, neuroregeneration, neuroprotection, experimental treatments, locomotor recovery

## Abstract

Spinal cord injury (SCI) is characterized by a complex and prolonged injury process that exacerbates the damage induced by the primary injury and inhibits the potential for regeneration. SCI frequently results in the devastating loss of neurological functions and thus has serious consequences on patient quality of life. Current treatments are limited and focus on early interventions for the acute management of complications. Therefore, the development of novel treatments targeting ongoing injury processes is required to improve SCI outcomes. We aimed to systematically review studies published in the last 10 years that examined experimental treatments with neuroregenerative and neuroprotective capabilities for the improvement of SCI. We analyzed treatments from 44 studies that were identified through a systematic literature search using three databases: PubMed, Web of Science and EMBASE (searched through Ovid). We performed a meta-analysis for Basso-Beattie-Bresnahan (BBB) locomotion test data and collected immunohistochemistry results to demonstrate neuroregenerative and neuroprotective properties of the treatments, respectively. The two treatments that illustrated the most significant improvements in functional recovery using the BBB test were the combined use of tetrahedral framework nucleic acid (tFNA) with neural stem cells (NSCs) and Fortasyn^®^ Connect (FC) supplementation. Both treatments also attenuated secondary injury processes as demonstrated through immunohistochemistry. Combined tFNA with NSCs and FC supplementation are promising treatments for the improvement of SCI as they both demonstrate neuroregenerative and neuroprotective properties. Further pre-clinical testing is required to validate and determine the long-term efficacies of these treatments for the improvement of SCI.

## 1. Introduction

Spinal cord injury (SCI) is characterized by damage to the spinal cord resulting in short- or long-term implications to normal sensory, motor, and autonomic functions thus leading to debilitating conditions through impacting an individual’s physical, psychological, and social well-being [1,2,3]. SCI can be due to traumatic or non-traumatic injuries, with traumatic SCI contributing to more than 90% of the total number of affected individuals [1,4]. Common causes for traumatic SCI include road traffic accidents, falls, sports, violence, whilst common causes for non-traumatic injuries include infection, cancer, and disease [1,2]. Globally, it is estimated that SCI affects between 250,000 to 500,000 individuals per year resulting in significant financial burdens for both patients and healthcare services [3,4,5,6,7]. The outcome of SCI depends heavily on the severity and the level of injury, with cervical level injuries, the most common type of SCI, typically resulting in quadriplegia, and thoracic level injures resulting in paraplegia [4]. Ultimately, SCI results in disruption to the complex multicellular interactions present within normal spinal cord physiology which consequently leads to compromised recovery [5,6].

Currently, treatments for SCI are relatively limited and focus on early diagnosis and surgical intervention during the immediate and primary phase of injury to limit the potential loss of neurological functions. However, ongoing injury processes such as the loss of neuronal and glial cells, demyelination, glial scar and cystic cavity formation observed during the secondary phase of injury, result in progressive neurodegeneration and inhibit the potential for regeneration to occur [5,6,7,8]. No reparative treatment exists for SCI with only Lyrica being approved as a palliative treatment for neuropathic pain. Therefore, there is an urgent medical need to devise new therapies that target the underlying pathophysiological processes after SCI.

There have been a number of experimental treatment strategies targeting these processes, for example to promote neuroprotection and neuroregeneration in order to improve the neurological outcomes following SCI. The aim of this study was to systematically review the experimental treatment options which have demonstrated, specifically, neuroregenerative and neuroprotective capabilities for SCI, developed in the last 10 years, and ascertain the best treatment options for further pre-clinical validation. Neuroregeneration through enhanced functional recovery, as observed using the Basso-Beattie-Bresnahan (BBB) locomotion test and neuroprotection through observing improvements in the reduction or inhibition of secondary injury processes using immunohistochemistry were the primary and secondary outcomes of this review and were used to determine the best treatment options for SCI developed in the last 10 years.

## 2. Materials and Methods

### 2.1. Review Process

A systematic review of the literature was performed by two authors (F.K. and Z.A.) whilst following the guidelines outlined by Preferred Reporting Items for Systematic Reviews and Meta-Analyses (PRISMA) [9]. The protocol for this systematic review is also published in the Open Science Forum registries and can be accessed using the following https://doi.org/10.17605/OSF.IO/PKQZ9.

### 2.2. Literature Search

A search of published literature that examined the effects of experimental treatments for SCI was carried out on 2nd February 2022 using three electronic databases: PubMed, Web of Science and EMBASE (searched through Ovid). The common search strings used for all three electronic databases were ‘(spinal cord injury) AND (treatments) AND (neuroprotection) AND (axon regeneration)’. Additional search terms included “AND/OR SCI” AND/OR neural regeneration”, “AND/OR neurorehabilitation”, “AND/OR rehabilitation” and “AND/OR neuroprotective”. The literature search was limited to reports written in the English language and only covering reports on neuroregenerative and neuroprotective treatments. Any duplicates generated amongst the three electronic databases were removed and initial shortlisting of manuscripts was performed through screening the titles and abstracts of papers against the outlined inclusion and exclusion criteria. Shortlisted studies were then reviewed and assessed for eligibility after a full text read which resulted in the final selection of 44 studies.

### 2.3. Inclusion and Exclusion Criteria

To determine eligibility for this systematic review, studies were screened against the following criteria: (1) only manuscripts describing preclinical in vivo animal studies were included (some of these studies also reported in vitro data but this was not analyzed nor included in our systematic review as we were only interested in studies that reported animal data and hence were closer to translation); (2) only studies published in the last 10 years from the years 2012 to 2022 were included; and (3) only studies examining effects of therapeutic treatments on cervical or thoracic spinal cord injuries were included, studies with other central nervous system injuries were not included. Clinical studies, reviews, systematic reviews, books/documents, clinical trials, meta-analyses, and conference articles were all excluded. Assessment of study eligibility was carried out in an unblinded and standardized manner and independently by two reviewers (F.I.K. and Z.A.).

### 2.4. Risk of Publication Bias

To evaluate the risk of bias in the final selection of the 44 studies, SYRCLE’s risk of bias tool for animal studies, adapted from Cochrane’s risk of bias tool was used [10]. Each study was assessed against 10 different risk parameters to determine the overall risk of bias, and responses to each risk parameter were recorded as the following: (a) yes, if the parameter was positively reported with sufficient supporting methodology; (b) yes, if the parameter was positively reported but had unknown or insufficient methodology; or (c) no, if the parameter was negatively reported in the study or if the parameter was not reported at all. The risk of bias was assessed by two independent reviewers (F.I.K. and Z.A), with any disagreements being settled through discussion.

### 2.5. Data Extraction and Synthesis

Data collection of the key characteristics of each paper including the author(s), year of publication, location of where the study was conducted, whether an in vitro or in vivo study was performed, the spinal level at which the injury was induced, the treatment used, and the experimental techniques performed were summarized in a pre-designed table. Data for the BBB locomotion test was also collected and summarized in a pre-designed table, studies that did not perform this test were identified and data was recorded as not reported. Data collection for immunohistochemical markers demonstrating neuroprotective or neuroregenerative effects of treatments was also reported and summarized within a pre-designed table, studies that did not perform immunohistochemistry were identified and recorded as not reported.

### 2.6. Statistical Analysis

Assessment of heterogeneity was performed by examining the differences across studies for methodological heterogeneity. We used Review Manager (RevMan 5.3, Cochrane Informatics & Technology, London, UK) to determine the Q and I^2^ statistics (in percentage) to establish variation between the studies attributed to heterogeneity. A meta-analysis of sub-groups of studies which reported overall improvement in BBB scores at different time points after injury and treatment was conducted in RevMan 5.3 (Cochrane Informatics & Technology), using the continuous data function, employing a random effects model and selecting mean difference as the effect measure.

## 3. Results

### 3.1. Study Selection

Applying the common search strings to all three electronic databases: PubMed, Web of Science and EMBASE, generated a total of 453 search results. After removing duplicates, 431 results remained for initial screening using the inclusion and exclusion criteria as previously outlined. 118 records were then excluded due to reasons not relevant to the study question such as being a non-experimental study which includes reviews, systematic reviews, books or documents, clinical trials, meta-analyses, and conference articles. In addition, any report not written in the English language was also excluded. This left 313 studies that were screened by reading the titles and abstracts following the previously outlined inclusion and exclusion criteria. Papers published outside the 10-year time period from 2012 to 2022 were excluded, papers assessing treatments for other central nervous system injuries other than SCI were also excluded, and papers attempting to enhance delivery of a previous known treatment or attempting to replicate the results of a previous treatment were also excluded, leaving 130 articles that required a full text read to assess for eligibility. Following full text reads of these articles, the final selection of 44 studies were identified and included for analysis in this systematic review [11,12,13,14,15,16,17,18,19,20,21,22,23,24,25,26,27,28,29,30,31,32,33,34,35,36,37,38,39,40,41,42,43,44,45,46,47,48,49,50,51,52,53,54]. The literature search process and screening for eligibility is presented in the PRISMA diagram below (Figure 1).

### 3.2. Study Characteristics

A summary of the key characteristics of each paper is presented in Table 1. The studies included in this review were conducted in a range of 16 different countries with a large proportion of the studies being based in China (47.7%) followed by the United States of America (USA) (9.1%), and Spain (6.8%). The effects of potential therapeutic treatments for SCI can be observed using both in vitro and in vivo methods. In this review, all of the 44 studies used in vivo methodology with a significant proportion of studies only using in vivo methodology (63.6%) and a smaller proportion utilizing both in vivo and in vitro.

The severity of the outcome of SCI can be influenced by the location of where the injury was obtained. The spinal level at which the injury was induced for each of the studies was therefore reported in Table 1. The majority of the studies used a thoracic level injury (93.2%) leaving only two studies by Karalija et al. [23] and Chen et al. [49] that performed cervical level injuries at the spinal levels C3 and C5, respectively. There was also a single study performed by Karalija et al. [24] that induced a SCI in the lumbar region of the spinal cord at L5. The most common injury was at the level T10 (25.0%) closely followed by T9 (22.7%) and then T8 (15.9%).

Eleven of the included studies had reported a range for the level of injury rather than a single spinal level and therefore, the range for all studies reporting thoracic level injuries was calculated to be from the level T6 to T12. The type of experimental treatment that was analyzed by the studies varied significantly as demonstrated in Table 1. Treatments ranged from pharmacological drugs, immunomodulatory treatments, dietary supplementations, stem cell-based transplantations, and even stem cell conditioned mediums. A proportion of the studies had also reported using combined treatment strategies for the synergistic treatment of SCI instead of a single experimental treatment.

A range of many different experimental techniques to analyze the effectiveness of the various treatments were performed in the 44 studies as listed in Table 1. A frequent key aim of these studies was to assess neuronal regeneration, and this was examined using several experimental procedures. An example includes the use of behavioral tests to observe improvements in the functional recovery of animals following SCI which thus indicates neuronal regeneration. A variety of different behavioral methods of testing were used such as the BBB (a 21-point scale based on the Basso 9-point Mouse Scale (BMS) adapted to rats) locomotion test, BMS scoring, inclined plane test, footprint analysis test, grid walking test, CatWalk gait test and sensory tests. The BBB locomotion test was the most commonly used behavioral test (72.7%) followed by BMS scoring (13.6%). As the BBB locomotion test was the most common behavioral test performed, the BBB test results reported in the studies was collected and analyzed in this review using a meta-analysis (see below). Neuronal regeneration was also examined using histological methods, this included the use of haematoxylin eosin (HE) staining to examine alterations/improvements in histology, and Nissl and cresyl violet staining to observe potential changes in neuronal numbers following treatment application.

A number of other experimental procedures were also performed to examine changes induced by the treatments and to therefore determine the effectiveness of the treatments for SCI. In particular, immunohistochemistry and other forms of immunostaining was performed in a large proportion of the studies (95.5%). Immunohistochemistry was also frequently coupled with Western Blotting to analyze changes in protein expression levels. Other experimental techniques carried out by the studies included terminal deoxynucleotidyl transferase biotin-dUTP nick end labelling (TUNEL) assays, enzyme-linked immunosorbent assay (ELISA), quantitative reverse transcription polymerase chain reaction (qRT-PCR), flow cytometry and electrophysiological tests.

### 3.3. The BBB Locomotion Test

The BBB locomotion test was a common behavioral test that was utilized by 32/44 of the studies. BBB tests were performed on treatment groups that were administered the experimental treatment and underwent SCI, and control groups that only underwent SCI and in some cases were administered a control medium such as phosphate-buffered saline (PBS). How often BBB tests were carried out varied amongst the studies with some studies performing daily BBB tests post SCI and some performing weekly BBB tests post SCI. In order to analyze the BBB test data, the final reported data value for control and treatment groups was collected for the 32 studies and presented in Table 2. The final timepoint for collected BBB test data varied significantly from 7 days (D7) post SCI to 294 days (D297) post SCI. Common timepoints such as D28, D35, D42 and D56 as well as studies analyzing the effects of combined treatments were sub-grouped for five individual meta-analyses, and a final meta-analysis was also performed on all of the 32 studies grouped together (see below).

Almost all studies reported an improvement in BBB scores in the treatment groups compared to controls apart from the studies by Liang et al. [34], Mountney et al. [17], and Machova-Urdzikova et al. [37]. Although the BBB score did not surpass control groups that only underwent SCI and no treatment, Liang et al. [34] did report a 5-point improvement in the BBB score for the treatment neural stem cell conditioned medium (NSCM) group compared to the second treatment group which used a control medium. Thus, indicating NSCM treatment to improve functional recovery greater than the control medium [34].

Mountney et al., 2013 [17] reported the same BBB score for the combined chondroitinase ABC (chABC) and sialidase treatment group and the control group. However, there was a 3-point increase in the BBB score for the sialidase alone treatment group compared to the control group [17]. This suggested that sialidase treatment improved functional recovery and that combination treatment of chABC and sialidase failed to do so [17].

In the study by Machova-Urdzikova et al. [37], there was no significant difference reported between the BBB scores of the epigallocatechin gallate (EGCG) treatment group and control group receiving saline treatment, which therefore suggested EGCG to have no significant effect on enhancing functional recovery. However, other behavioural tests such as the flat beam test that assesses balance and coordination skills were also performed in this study [37]. Results from this test reported a significant decrease in the time needed to cross the beam in the EGCG treated group compared to the control group, and therefore suggested an improvement in the behavioural outcomes following SCI with EGCG treatment [37].

#### BBB Meta-Analysis

The final experimental timepoint reported for BBB test data varied amongst the 32 studies; therefore, if three or more studies reported the same end timepoint they were sub-grouped together for a meta-analysis. As a result, a meta-analysis was performed on studies with the timepoints: D28, D35, D42 and D56 (Figure 2, Figure 3, Figure 4 and Figure 5). A meta-analysis was also performed on studies that used combined treatments (Figure 6) and a final meta-analysis was carried out on all 32 studies that performed the BBB locomotion test (Figure 7).

There were nine out of the 32 studies that reported their final recorded BBB data value 28 days following SCI [11,13,25,27,30,31,33,34,54]. A meta-analysis was performed on these studies and the data is presented in Figure 2. Study heterogeneity was analyzed using the I2 statistic. For the D28 sub-grouped studies, a 100% heterogeneity between the studies was reported indicating very high heterogeneity. The meta-analysis also demonstrated that in all studies apart from the study performed by Liang et al. [34], the treatment groups had higher BBB scores, recorded at D28 post SCI, that were statistically significant compared to control groups. Therefore, suggesting these treatments to enhance.

Only three out of the 32 studies reported their final BBB test score 35 days after SCI [17,28,41]. These studies were sub-grouped for a meta-analysis as presented in Figure 3. As Mountney et al. [17] reported no difference in the BBB score for the combined chABC and sialidase treatment group, the BBB score for the sialidase alone treatment group was used for this meta-analysis instead. The heterogeneity between the studies was very high at 97% as indicated by the I^2^ statistic. The meta-analysis also demonstrated that all three studies reported a statistically significant increase in the BBB score for the treatment groups compared to controls. Thus, indicating the treatments to improve functional recovery greater than controls following SCI. The overall effect of these treatments was determined to be statistically significant as indicated by the Z statistic and the black diamond in the forest plot, with a mean difference of 3.22 [1.45, 5.00], *p* < 0.00001.

There was 6 out of the 32 studies that reported the final BBB test score 42 days following SCI [19,32,46,48,51,53]. These studies were subsequently sub-grouped for a meta-analysis (Figure 4). The heterogeneity between these studies was 93% and thus very high as calculated by the I^2^ statistic. All of these studies had higher BBB scores that were statistically significant for the treatment groups at 42 days following SCI compared to control groups, which therefore indicated a greater improvement in functional recovery following SCI that was due to the treatments applied. The overall effect of all the treatments was a statistically significant increase in the BBB score compared to control groups as indicated by the Z statistic and black diamond in the forest plot, with a mean difference of 4.00 [2.54, 5.46], *p* < 0.00001.

There were seven out of the 32 studies that recorded the final BBB test score 56 days after SCI [15,18,26,29,40,47,50]. These studies were sub-grouped for a meta-analysis as presented in Figure 5. The I2 statistic reported a very high heterogeneity between these studies at 99%. All of these studies reported a higher BBB score for the treatment groups compared to control groups which therefore indicated enhanced functional recovery that was due to the treatments administered. However, this was only statistically significant in 6 out of the 7 studies [15,26,29,40,47,50]. The meta-analysis demonstrated that although the treatment resulted in a higher BBB score compared to controls, in the study by Maqueda et al. [18], this increase in the BBB score was not statistically significant as indicated by the 95% confidence interval (CI) in the forest plot. The overall effect of the increase in BBB scores by all the treatments, however, was found to be statistically significant compared to the controls as indicated by the Z statistic and black diamond in the forest plot, with a mean difference of 2.15 [0.77, 3.53], *p* < 0.002.

From the 32 studies that performed the BBB test, 9 of them analyzed a combined treatment for recovery of SCI [15,17,22,26,27,28,40,47,50]. These studies were sub-grouped for a meta-analysis of their final recorded BBB test score (Figure 6). There was a 99% heterogeneity between these studies as calculated by the I^2^ statistic. All of the studies, apart from the study by Mountney et al. [17] which investigated the use of combined chABC and sialidase treatment, had an increase in the BBB score in the treatment groups compared to controls, which therefore suggested improved functional recovery. However, only 7 of the studies were statistically significant [15,26,27,28,40,47,50]. The study by Bonilla et al. [22] did not demonstrate a statistically significant increase in the BBB score of the treatment group as indicated by the CI in the forest plot. Despite this, the overall effect of the treatments in all 9 of these studies was a statistically significant increase in BBB score compared to control groups as indicated by the black diamond in the forest plot and Z statistic, with a mean difference of 3.22 [1.59, 4.86], *p* < 0.0001.

A meta-analysis was performed on the final timepoints for recorded BBB scores from all 32 studies that performed the BBB test [11,12,13,14,15,16,17,18,19,22,25,26,27,28,29,30,31,32,33,34,37,40,41,43,45,46,47,48,50,51,53,54]. The heterogeneity between studies was 100% as calculated by the I^2^ statistic. Out of the 32 studies, 29 studies demonstrated a statistically significant increase in the BBB scores for the treatment compared to control groups, thus suggesting these treatments to enhance functional recovery [11,12,13,14,15,16,18,19,22,25,26,27,28,29,30,31,32,33,40,41,43,45,46,47,48,50,51,53,54]. The study by Mountney et al. [17] showed no difference in the BBB score for the combined treatment and control group. Despite this, Mountney et al. [17] did demonstrate a statistically significant increase in the BBB score for the sialidase alone treatment compared to the control group as previously described and presented in Figure 3. This therefore indicated sialidase treatment to enhance functional recovery greater than the combined use of chABC and sialidase [17]. The studies by Liang et al. [34] and Machova-Urdzikova et al. [37] also did not result in an increase in the BBB score for the treatment groups, instead the controls were favored. Despite this, the overall effect for all of the treatments in the 32 studies was a statistically significant increase in the BBB score compared to control groups as demonstrated by the black diamond in the forest plot and the Z statistic with a mean difference of 3.86 [1.36, 6.36], *p* < 0.002. 

### 3.4. Immunohistochemistry

Experimental techniques such as immunohistochemistry and other forms of immunostaining were frequently performed in all of the 44 studies apart from two studies by Caglar et al. [11] and Li et al. [48]. Immunohistochemistry was performed on both control groups and treatment groups in vivo in order to compare and analyze differences in the expression of markers that were due to the treatments administered. These results from immunohistochemical staining would therefore allow for the determination of whether treatments provided neuroprotection and/or whether treatments were able to induce neuroregeneration following SCI. We collected the results for common immunohistochemical markers that were used in the 42 studies that performed immunohistochemistry and presented the results in Table 3.

A frequent marker that was observed in many of the studies was glial fibrillary acidic protein (GFAP) as demonstrated in Table 3. GFAP is commonly expressed in astrocytes and is associated with reactive astrogliosis [12]. Reactive astrogliosis induces the formation of the glial scar and therefore inhibits the potential for regeneration to occur [18]. Consequently, a reduction in GFAP staining is a desired outcome as it suggests a reduction or inhibition of reactive astrogliosis [18]. A large proportion of the studies stained for GFAP and in many of these studies GFAP staining was decreased in the treatment groups compared to controls which therefore indicated neuroprotective properties of the treatments due to the reduction or inhibition of reactive astrogliosis [12,15,19,22,25,29,30,35,38,39,40,41,42,50,53].

Neurofilament (NF) or neurofilament 200 (NF-200) are markers for neurons and are common markers used to observe potential neuronal/axonal growth [24,27]. An increase in the staining of this marker therefore suggests neuroregeneration through neuronal or axonal regrowth and spouting [27]. As demonstrated in Table 3, a significant proportion of the studies used NF or NF200 as markers for in vivo immunohistochemical tests on control and treatment groups following SCI. The majority of these studies reported an increase in NF or NF-200 staining in the treatment groups, which thus indicated possible neuroregeneration that was induced by the treatments used [13,14,16,18,19,24,27,28,30,31,35,36,38,47,51,53]. There was only one study by Chen et al. [49] that did not find any difference in NF200 staining in the treatment or control groups however, they did report an increase in NeuN staining. This is another marker for neurons.

Another marker that was frequently stained for using immunohistochemistry was ionized calcium adaptor molecule 1 (Iba-1). This is a microglia marker and staining for this marker allows for the identification of microglia activation [12]. The activation of microglia contributes towards the neuroinflammatory environment and thus further potentiates secondary injury processes such as glial scar formation [41]. Inhibiting or reducing the levels of microglia activation would therefore limit the damaging neuroinflammatory response as well as other secondary injury events [41]. As presented in Table 3, many of the studies stained for the marker Iba-1 and found a decrease in Iba-1 staining in the treatment groups compared to control groups, which suggested that the treatments used provided neuroprotection following SCI through inhibiting microglia activation or reducing the levels of microglia activity [12,18,25,29,35,38,41,43]. Aleksić et al. [44] also found decreased Iba-1 staining in the treatment group however, this was not statistically significant compared to the control group. Furthermore, another study by Bimbova et al. [53] also found decreased Iba-1 staining in the treatment compared to control group at 24 h following SCI, however this difference in staining was no longer observed six weeks following injury, which therefore suggested that the treatment had short term effects on microglia activation.

The survival of neurons and neuronal growth or sprouting were recurrent outcomes assessed by many of the studies by using immunohistochemistry to stain for markers of mature neurons. This allowed for the effects of treatments on changes in neuronal levels or neuronal sprouting/regrowth to be observed. Neuronal nuclear protein (NeuN) and serotonin, which is also known as 5-hydroxytryptamine (5-HT), were two markers that were frequently used in many of the studies in order to assess these outcomes. The majority of the studies that stained for NeuN, found increased NeuN staining in the treatment groups in comparison to controls [15,19,22,25,34,36,42,45,49,50]. This suggested that these treatments promoted the survival of neurons as less NeuN-positive neurons were found in the control groups and thus illustrated the neuroprotective properties of these treatments. However, there was one study by Aleksić et al. [44] that found no difference in NeuN staining in the treatment and control groups, which suggested that the treatment used did not affect the survival of neurons following SCI. Similarly, 5-HT was another useful marker used to observe any possible effects of the treatments on neurons following SCI. The majority of the studies that stained for 5-HT found increased 5-HT staining in the treatment groups compared to control groups which indicated the treatments to induce regeneration of 5-HT positive neurons [16,23,38,42,46,52]. There was only one study by Mountney et al. [17] that found no difference in staining of 5-HT between treatment and control groups which therefore suggested the treatment to have no effect on 5-HT positive neurons.

Caspase-3 is a caspase protein that is involved in the apoptotic pathway and as the apoptotic loss of neuronal and glial cells is frequently observed during SCI, caspase-3, or cleaved caspase-3 (C-caspase-3) was used as a marker for immunohistochemistry in some studies as presented in Table 3 [13,29,34,53]. Studies that used this marker aimed to determine whether the application of treatment would result in changes in the levels of apoptosis observed in mainly neuronal cells, but also in other cells such as oligodendrocytes. All of the studies that examined this marker found a decrease in caspase-3 or C-caspase-3 staining which indicated a reduction in the apoptotic loss of neuronal cells and thus demonstrated the neuroprotective properties of the treatments used [13,29,34,53].

### 3.5. Risk of Bias

All 44 studies included in this review were assessed for the risk of publication bias using SYRCLE’s risk of bias tool for animal studies adapted from Cochrane’s risk of bias tool [10]. The summary analysis of the risk of bias for all 44 studies is presented below in Figure 8 and the results for each individual studies is presented in Figure 9.

All of the studies stated their primary outcomes and described their baseline characteristics however, none of the studies reported using a sample size calculation to determine the appropriate number of animals needed for the study. A proportion of the studies (22.7%) outlined the incomplete outcome data which was largely involving the loss of animals following SCI or in some cases animals that did not demonstrate sufficient functional loss after SCI and thus were excluded from the study. A large proportion of studies (68.2%) blinded the assessors and examiners to the experimental conditions in order to prevent experimental bias however, a significant number of studies (95.5%) failed to randomize outcome assessments leaving only two studies by Mountney et al. [17] and Maqueda et al. [18] that did randomize outcome assessments. The blinding of participants and personnel was also only positively reported in two studies by Pan et al. [42] and Aleksić et al. [44]. There were no studies that reported random housing for the animals used and only a small proportion of studies (9.1%) concealed the allocation of the animals to each experimental group. Despite this, random sequence generation to allocate animals to their experimental group was applied to a large proportion of the studies (65.9%). To summarize, the risk of bias for the studies overall was relatively high as there was a significant proportion of the studies that failed to outline many of the risk parameters listed in Figure 8 and Figure 9.

## 4. Discussion

The aim of this report was to systematically review potential experimental treatment options that demonstrated neuroprotective and neuroregenerative capabilities following SCI in order to determine the current best treatment options to undergo further pre-clinical testing for validation. To do this, we searched for papers published in the last 10 years using three scientific databases: PubMed, Web of Science and Ovid Embase. The same relevant search terms were applied to each database resulting in a total of 453 search results. These papers were screened against an inclusion and exclusion criteria and assessed for eligibility to narrow down the results to the final 44 studies that were included in this review [11,12,13,14,15,16,17,18,19,20,21,22,23,24,25,26,27,28,29,30,31,32,33,34,35,36,37,38,39,40,41,42,43,44,45,46,47,48,49,50,51,52,53,54]. The details of the key characteristics of all 44 studies were collected and summarized which allowed for the identification of the BBB locomotion test and immunohistochemistry as two common experimental procedures performed amongst the 44 studies. Thus, neuroregeneration through enhancing functional recovery analyzed using the BBB locomotion test, and neuroprotection through observing changes induced by treatments using immunohistochemical markers were the primary and secondary outcomes assessed in this review in order to determine the best treatment options for SCI. Following a meta-analysis of the data from the BBB locomotion test and collection of immunohistochemistry results, we identified multiple studies with experimental treatments that enhanced functional recovery and attenuated secondary injury processes. We also determined a relatively high risk of bias amongst the 44 studies following analysis using SYRCLE’s risk of bias tool for animal studies adapted from Cochrane’s risk of bias tool [10]. The results from this systematic review have therefore highlighted several studies with promising treatment options for the improvement of SCI through illustrating neuroregenerative and/or neuroprotective capabilities.

### 4.1. Neuroregeneration

The loss of normal sensory, motor, and autonomic functions due to the degenerative injury process is frequently observed following SCI, and therefore the ability to induce regenerative effects in order to improve neurological outcomes is a desirable feature for many experimental treatments [1,2,3]. Neuroregeneration can be demonstrated through the use of behavioral tests that assess the improvement in the functional recovery of animals following SCI. In this review, almost all of the studies (90.9%) used one or more behavioral tests to examine improvements in functional recovery induced by the treatments administered.

The BBB locomotion test was the most common behavioral test utilized amongst the studies (72.7%) and so data from each study was collected and presented in Table 2. The time period for how often BBB tests were performed following SCI varied significantly, with a large proportion of studies performing tests for >28 days following SCI and only three studies by Wang et al. [16], Hou et al. [12] and Wang et al. [45] performing BBB tests for <28 days following SCI. Consequently, only the final reported BBB test score for the treatment and control groups from each study was collected, as this allowed for a fair comparison of the improvement in functional recovery between the studies.

BBB data for the common time periods: D28, D35, D42, D56 and data for combined treatments was then sub-grouped for five different meta-analyses, and a final meta-analysis was also performed for the all the studies that used the BBB test. As illustrated by the meta-analyses, a large proportion of studies had shown their experimental treatments to significantly increase the BBB scores compared to controls and thus demonstrated the use of promising treatments that enhance functional recovery following SCI. There were only two studies by Liang et al. [34] and Machova-Urdzikova et al. [37] that failed to increase the BBB score in the treatment groups.

In the study by Liang et al. [34], neural stem cells were cultured and then removed from the culture medium leaving a neural stem cell conditioned medium (NSCM) that was used to determine whether it would be an effective treatment for SCI. The results from the BBB test and meta-analysis reported the control group achieved a final score of 18.9 and the NSCM group to achieve a final score of 16.2 [34]. However, Liang et al. [34] also investigated the use of a control medium, and this group achieved a final score of 10.1. It was therefore concluded by Liang et al. [34] that NSCM treatment resulted in a five-point improvement in the BBB score when compared to the control medium group. Nevertheless, as illustrated by the meta-analysis in this review, NSCM treatment failed to enhance functional recovery greater than sham controls that only underwent a SCI and thus failed to demonstrate neuroregenerative properties.

The study by Machova-Urdzikova et al. [37] also reported a lower BBB score of 6.6 for the EGCG treated group compared to 6.8 for the control group however, this was not reported to be a significant difference. Instead, neuroregeneration was demonstrated through other behavioral tests such as the flat beam test where EGCG treated animals required significantly less amount of time to cross the beam compared to the control group, which therefore suggested EGCG as a promising treatment for the improvement of SCI [37]. On the other hand, the three studies that resulted in the most significant differences in BBB scores between the treatment and control groups and thus demonstrated the most promising results for the improvement in functional recovery, were the studies by Caglar et al. [11], Ma et al. [15] and Pallier et al. [25] with mean differences of 17.00, 8.17 and 8.00, respectively, as illustrated in the meta-analysis.

Caglar et al. [11] investigated the use of the pharmaceutical drug Riluzole as a treatment for SCI. The results from the BBB tests suggested Riluzole treatment to improve functional recovery significantly, as the BBB score reported at D28 in the treatment group was recorded as 19.0 compared to 2.0 in the control group, producing a mean difference of 17.00 as presented in the meta-analysis [11]. However, this result was produced in an experimental group (Group 8) where Riluzole was administered every 12 h for 5 days before the SCI was induced and so does not reflect normal conditions for the treatment of SCI [11]. Caglar et al. [11] investigated several experimental groups that tested Riluzole treatment at different time points before injury. There was only one experimental group (Group 4) where Riluzole was administered every 12 h for 7 days following injury, and in this group the BBB score at D28 was recorded as 12.0, which is still significantly higher than the control group [11]. Despite this, it was Group 8 that demonstrated the most significant improvement for SCI, and it was concluded that Riluzole treatment is therefore most beneficial prior injury for pre-operative patients at high risk of neurological injury [11]. However, as Riluzole treatment in Group 4 still produced positive results for the improvement of functional recovery following SCI, Riluzole may still prove to be a promising treatment. Additional research into the use of Riluzole after SCI is needed to further investigate potential beneficial effects of Riluzole treatment.

The study by Ma et al. [15] produced the next highest mean difference in BBB scores between the treatment and control groups as illustrated by the meta-analysis. The BBB score at D56 in the treatment group was recorded as 18.67 compared to 10.50 in the control group producing a mean difference if 8.17 [15]. In this study, a combined treatment of tetrahedral framework nucleic acid (tFNA) and neural stem cells (NSCs) was used to investigate the effects on the treatment of SCI [15]. Ma et al. [15] also investigated the use of both treatments, tFNA and NSCs, individually in order to determine whether the co-transplantation of tFNA with NSCs would enhance SCI recovery greater than either treatment alone. The results for the BBB scores at D56 for individual treatments of tFNA and NSCs were 15.33 and 15.83, respectively; thus, proving the synergistic treatment of tFNA and NSCs to enhance functional recovery more than either treatment alone [15]. The combined use of tFNA and NSCs for the treatment of SCI should therefore be investigated further as it has demonstrated promising results for the improvement of functional recovery following SCI.

Pallier et al., 2015 [25] had the next highest mean difference in BBB scores as demonstrated by the meta-analysis. The BBB score for treatment group at D28 was recorded as 17.00 and 9.00 for the control group producing a mean difference of 8.00 [25]. This study investigated the use of dietary supplementation using Fortasyn^®^ Connect (FC) as a treatment for SCI [25]. FC is a combination of multiple nutrients such as docosahexaenoic acid (DHA), eicosapentaenoic acid (EPA), phospholipids, monophosphates, choline, vitamins B12, B6, C and E and selenium [25]. As demonstrated by the BBB data and meta-analysis, FC treatment resulted in a significant improvement in functional recovery compared to the control group thus indicating FC as a promising supplementation treatment for further research into SCI treatments [25].

### 4.2. Neuroprotection

During the pathogenesis of SCI, there are many processes that exacerbate the outcome of the initial injury, such as the necrosis and apoptosis of neuronal and glial cells that can result in the loss of neurological functions, and the activation of astrocytes that produces the glial scar which in turn restricts the potential for regenerative processes to occur [5,8]. Targeting these injury processes would therefore aid in the recovery of SCI by providing neuroprotection. Consequently, many experimental treatments aim to provide neuroprotection by completely inhibiting or reducing these secondary injury processes. In this systematic review, immunohistochemistry and other forms of immunostaining were identified as common experimental procedures that were frequently performed in a significant proportion (95.5%) of the 44 studies. Thus, the results for common immunohistochemical markers were collected and summarized in Table 3. This allowed for changes in the staining of immunohistochemical markers to be observed, which therefore, demonstrated potential neuroprotective effects that were induced by the treatments administered.

As illustrated from the results in Table 3, there was a significant proportion of the studies that demonstrated treatments with neuroprotective capabilities. Many of these treatments may therefore be potentially effective and promising strategies for the improvement of SCI. However, only 30 of these studies performed both immunohistochemistry and the BBB locomotion test to demonstrate neuroprotective and neuroregenerative properties [12,13,14,15,16,17,18,19,22,25,26,27,28,29,30,31,32,33,34,37,40,41,43,45,46,47,50,51,53,54]. Out of these studies, there was only a few that failed to demonstrate neuroprotection alongside neuroregeneration. For example, in the study by Wang et al. [16] Quercetin was investigated as a potential treatment for SCI. Quercetin treatment successfully managed to enhance functional recovery greater than controls as demonstrated in the meta-analysis however, the treatment also resulted in an increase in GFAP staining which indicated increased astrocyte activation [16]. Wang et al. [16] concluded that this was protective as activation of astrocytes enhances recovery during the early stages following SCI. Despite this, it is important to consider the long-term degenerative consequences of astrocyte activation such as glial scar formation which would ultimately restrict regeneration from occurring. Consequently, although Quercetin treatment may enhance functional recovery, further research to improve the treatment and modulate its effects on astrocyte activation is still required.

On the other hand, out of the three studies that resulted in the greatest improvements in functional recovery using the BBB test, only two of the studies by Ma et al. [15] and Pallier et al. [25] demonstrated neuroprotection using immunohistochemistry. Instead, Caglar et al. [11] demonstrated neuroprotection by examining histopathological changes through the use of haematoxylin eosin (HE) staining. The study found that Riluzole treatment resulted in a greater median number of neurons and a reduction in the number of glial cells compared to controls, which therefore illustrated the neuroprotective properties of Riluzole [11]. However, similar to the BBB test, these results were produced from Riluzole treatment before SCI and thus suggests Riluzole to have greater efficacy when administered prior injury [11].

Immunohistochemical results from Ma et al. [15] successfully demonstrated the neuroprotective properties of the combined treatment of tFNA and NSCs. The markers Nestin and NeuN were increased in the combined treatment group which suggested the treatment to increase the survival and proliferation of NSCs, as well as the differentiation of NSCs into neurons, and thus demonstrated the neuroprotective properties but also neuroregenerative properties of combined tFNA and NSCs treatment [15]. There was also decrease in GFAP staining which indicated the combined treatment to reduce NSC differentiation into astrocytes and thus reduce the formation of the glial scar [15]. Overall, the combined treatment of tFNA and NSCs has shown effective results for the improvement of SCI through demonstrating neuroprotective properties by increasing the survival of neurons and reducing glial scar size, as well as demonstrating neuroregenerative properties through the significant improvements in functional recovery [15]. Further research into the synergistic treatment of tFNA and NSCs would thus be a promising area for the development of SCI treatments.

Similarly, Pallier et al. [25] has also reported positive results for the immunohistochemical tests performed. Increased NeuN staining was observed in the FC treated group which suggested the treatment to reduce neuronal loss [25]. GFAP staining was decreased which indicated FC treatment to reduce reactive astrogliosis and thus reduce the formation of the glial scar [25]. Other neuroprotective properties were also demonstrated through the reduction in Iba-1 staining which indicated a reduction in microglia activation and thus neuroinflammation as well as increased staining for oligodendrocytes in the FC treated group [25]. As a result, FC treatment has demonstrated both neuroprotective and neuroregenerative properties and is therefore another promising area for further research into treatments for SCI.

### 4.3. Risk of Bias Analysis

In this systematic review, we analyzed the risk of bias for the 44 included studies by assessing each study against 10 different risk parameters using SYRCLE’s risk of bias tool for animal studies adapted from Cochrane’s risk of bias tool [10]. Through this assessment, it was determined that the risk of bias for the 44 studies was relatively high as many studies failed to outline several of the risk parameters. In particular, randomization was identified to be a significant issue in many of the studies.

Out of the 44 studies, 65.9% of them positively reported random sequence generation to allocate animals to the treatment and control groups and only a very small proportion of the studies (9.1%) concealed this allocation process, thus giving rise to a high selection bias despite the baseline characteristics being positively reported in all 44 of the studies [10]. Furthermore, none of the 44 studies reported random housing for animals and only two studies by Pan et al. [42] and Aleksić et al. [44] reported the blinding of participants and personnel. The lack of both of these parameters being positively reported in the studies suggests possible high levels of performance bias within the studies [10]. Blinding of the outcome assessment was positively reported in 68.2% of the studies leaving 13 studies that failed to do so and thus increasing the chances of detection bias within these studies [10]. However, despite 68.2% of the studies blinding the outcome assessors, only two studies by Mountney et al. [17] and Maqueda et al. [18] positively reported randomization for the outcome assessments. Therefore, increasing the chances of a high detection bias in the majority of the 44 studies [10].

A small proportion of the studies (22.7%) described their incomplete outcome data of animals that were excluded from the experiments. This was largely involving animals that did not demonstrate sufficient functional loss after SCI when analyzed using the BBB locomotion test, or animals that were lost due to the SCI procedure itself. There were no studies that reported using sample size calculations to determine the necessary number of animals per study and so the impact of smaller sample sizes should be considered for the validity of each study [10]. The primary outcomes on the other hand, were specified in all of the 44 studies. For future studies, addressing and outlining the risk parameters described in SYRCLE’s risk of bias tool for animal studies adapted from Cochrane’s risk of bias tool is recommended to reduce a high risk of bias [10]. In addition, the high risk of bias in animal studies may be also avoided by stricter adherence to a set of standardized techniques in animal experiments based on the ARRIVE guidelines (Animal Research: Reporting of In Vivo Experiments) [55].

### 4.4. Limitations

A limitation of this systematic review is the use of only the BBB locomotion test to analyze improvements in functional recovery. The BBB locomotion test was identified as the most frequent behavioral test used amongst the 44 studies (72.7%) and so data was collected for a meta-analysis leaving 12 out of the total 44 studies that were not included in the meta-analysis [20,21,23,24,35,36,38,39,42,44,49,52]. Some of these studies may have utilized other methods to analyze improvements in functional recovery such as the ladder crossing or inclined plane test. Consequently, any significant improvements in functional recovery induced by these treatments were not analyzed in this review. Another limitation regarding the BBB locomotion test involves the range of different end time points for the last recorded BBB score which impacted how well comparisons between studies were made. To resolve this, we sub-grouped common end time points of D28, D35, D42 and D56 to clearly illustrate the differences in improvements in functional recovery between studies that used the same end time point. The use of only immunohistochemistry to demonstrate the neuroprotective capabilities of treatments is also another limitation of this review. There may have been studies that performed other experimental methods such as histology analysis using, HE, cresyl violet or Luxol fast blue staining to assess general and myelin-related changes that indicated neuroprotection and were induced by the treatments administered. Another limitation of this review is the high risk of bias for the 44 included studies that was determined by the SYRCLE’s risk of bias tool for animal studies adapted from Cochrane’s risk of bias tool [10]. Additionally, the literature search process also limits the range of papers included and analyzed in this review, as only three electronic databases were used and only papers that were written in the English language and published in the last 10 years (2012–2022) were included.

### 4.5. Future Studies

Since the most promising treatments for improving functional recovery after SCI were Riluzole, tFNA + NSC and Fotasyn Connect (FC) [11,15,25], we recommend further studies to evaluate these compounds in preclinical models prior to potential clinical evaluation. In fact, a multi-centre clinical trial was planned with Riluzole in SCI patients, the RISCIS study (NCT01597518), however, the study has been terminated due to slow enrollment of patients. FC is a dietary supplement containing a combination of nutrient precursors and co-factors known to be needed in the synthesis of neuronal membranes. FC includes docosahexaenoic acid, eicosapentaenoic acid, uridine-5’-mono-phosphate, choline, phospholipids, selenium, and B, C, and E vitamins. FC has been tested in clinical trials involving Alzheimer’s disease patients and shown to be beneficial in the early period after disease onset [56]. Tetrahedral framework nucleic acid (tFNA) and NSCs provides a promising future therapy where co-transplantation of biomaterials with NSCs could offer superior healing capacity after SCI. Previous studies have also shown that tFNAs promote NSC proliferation, migration and differentiation into neurons, whilst possessing ani-inflammatory and antioxidative effects on macrophages [57,58,59]. Therefore, this could represent a useful future therapeutic.

## 5. Conclusions

This systematic review has highlighted several studies, published in the last 10 years, that demonstrated effective experimental treatment options with neuroregenerative and neuroprotective properties for the improvement of SCI. In particular, the combined use of tFNA and NSCs and the use of FC dietary supplementation have shown to be the most promising treatments for SCI. Both treatments have shown neuroregenerative capabilities by significantly enhancing functional recovery greater than controls as demonstrated by the increase in BBB scores achieved in the treatment groups. Both treatments have also demonstrated neuroprotection through attenuating secondary injury processes that exacerbate the overall outcome of SCI. In particular, co-transplantation of tFNA with NSCs has shown to increase neuronal survival and differentiation of NSCs into neurons following SCI, and FC supplementation has demonstrated the ability to reduce degenerative injury processes such as reactive astrogliosis and neuroinflammation from occurring following SCI. However, further pre-clinical testing is still required in order to validate and determine the long-term efficacies of FC supplementation and combined tFNA and NSCs treatment on the improvement of SCI should discuss the results and how they can be interpreted from the perspective of previous studies and of the working hypotheses. The findings and their implications should be discussed in the broadest context possible. Future research directions may also be highlighted.

## Figures and Tables

**Figure 1 cells-11-03409-f001:**
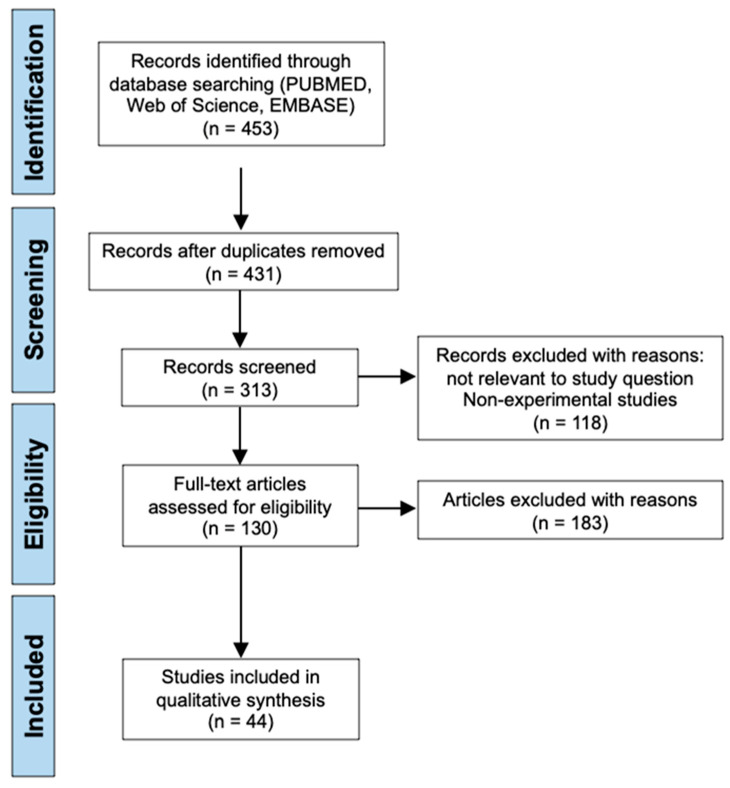
PRISMA flow chart illustrating the literature search process and screening of papers included in this systematic review.

**Figure 2 cells-11-03409-f002:**
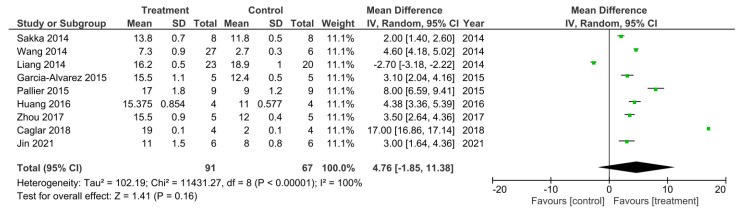
Meta-analysis for the effect of single treatments on the improvement of BBB scores analyzed at 28 days after spinal cord injury in nine studies [11,13,25,27,30,31,33,34,54].

**Figure 3 cells-11-03409-f003:**
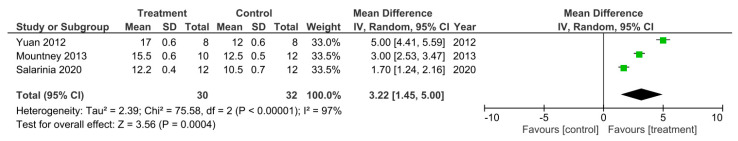
A meta-analysis for the effect of single treatments on the improvement of BBB scores analyzed at 35 days after spinal cord injury in three studies [17,28,41].

**Figure 4 cells-11-03409-f004:**
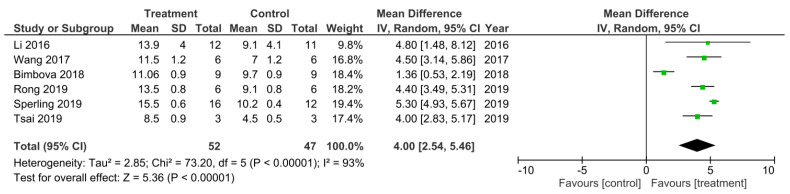
A meta-analysis for the effect of single treatments on the improvement of BBB scores analyzed at 42 days after spinal cord injury in six studies [19,32,46,48,51,53].

**Figure 5 cells-11-03409-f005:**
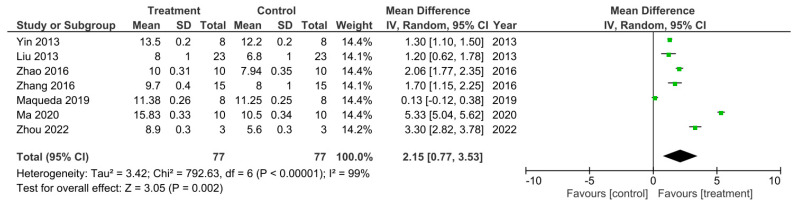
A meta-analysis for the effect of single treatments on the improvement of BBB scores analyzed at 56 days after spinal cord injury in seven studies [15,18,26,29,40,47,50].

**Figure 6 cells-11-03409-f006:**
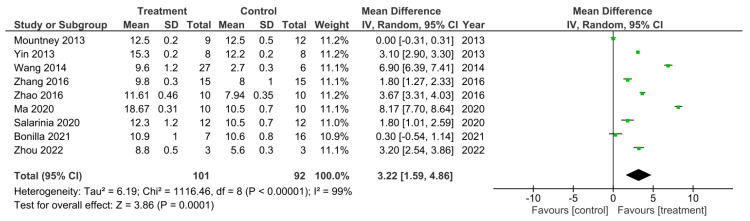
A meta-analysis for the effect of combined treatments on the improvement of BBB scores analyzed at the final recorded timepoints after spinal cord injury in nine studies [15,17,22,26,27,28,40,47,50].

**Figure 7 cells-11-03409-f007:**
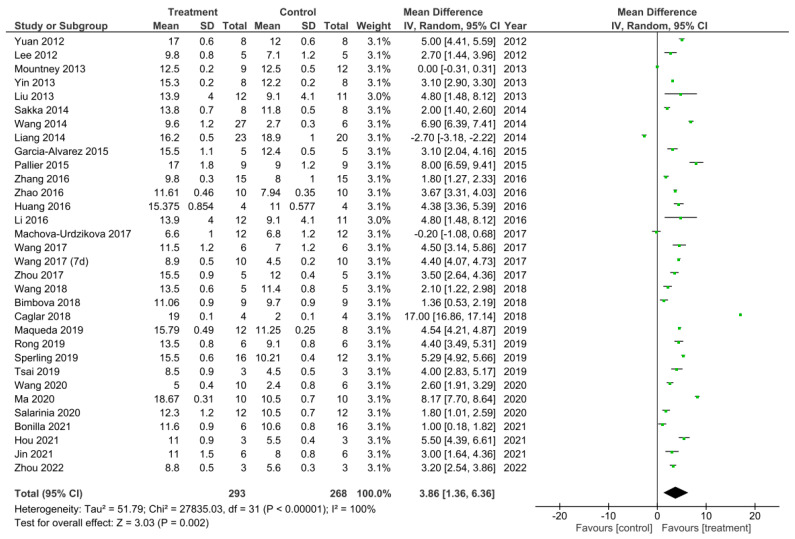
A meta-analysis for the effect of all treatments on the improvement of BBB scores analysed at the final recorded timepoints after spinal cord injury in all 32 studies that performed the BBB test [11,12,13,14,15,16,17,18,19,22,25,26,27,28,29,30,31,32,33,34,37,40,41,42,43,45,46,47,48,50,51,53,54].

**Figure 8 cells-11-03409-f008:**
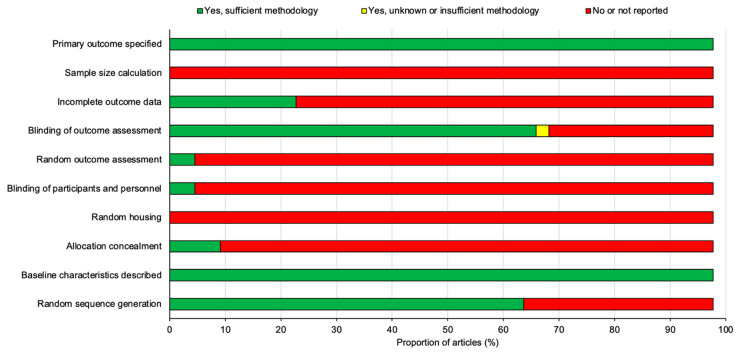
A summary diagram of the risk of bias for all 44 studies included in this review. All studies were assessed against ten different risk parameters as listed on the y-axis.

**Figure 9 cells-11-03409-f009:**
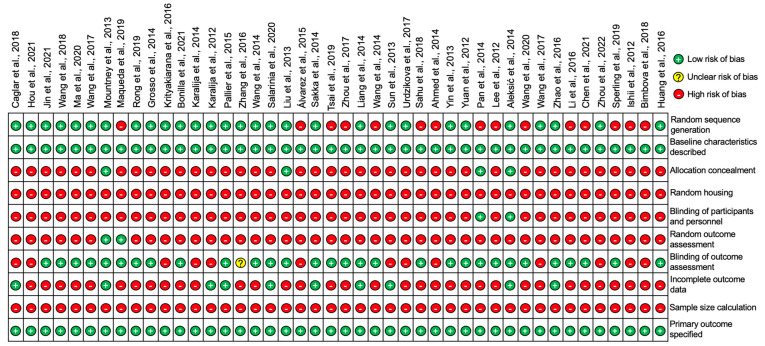
Diagram to represent the risk of bias in individual studies for all 44 studies included in this review [11,12,13,14,15,16,17,18,19,20,21,22,23,24,25,26,27,28,29,30,31,32,33,34,35,36,37,38,39,40,41,42,43,44,45,46,47,48,49,50,51,52,53,54].

**Table 1 cells-11-03409-t001:** Summary of the key characteristics of all 44 included studies.

Author	Location	In Vitro/ In Vivo	Level of Injury	Treatment	Experimental Techniques Performed
Caglar et al., 2018 [11]	Turkey	In vitro and in vivo	T7–T9	Riluzole	BBB locomotion test, HE staining, histology analysis
Hou et al., 2021 [12]	China	In vitro and in vivo	T9–T10	Tauroursodeoxycholic acid (TUDCA)	BBB locomotion test, footprint test, histology, immunofluorescence, Western Blot, TUNEL assay, Sholl analysis, HE staining
Jin et al., 2021 [13]	China	In vitro and in vivo	T9	Morin	BBB locomotion test, immunofluorescence, Western Blot, cytoxicity assay
Wang et al., 2018 [14]	China	In vitro and in vivo	T10	Endothelial progenitor cell-conditioned medium (EPC-CM)	BBB locomotion test, histology, immunohistochemistry, TUNEL assay, flow cytometry, HE staining, q-RT-PCR
Ma et al., 2020 [15]	China	In vivo	T9	Tetrahedral framework nucleic acid (tFNA) and neural stem cells (NSCs)	BBB locomotion test, cell proliferation assay (BrdU labelling), flow cytometry, TUNEL staining, HE staining, histology, immunohistochemistry
Wang et al., 2017 [16]	China	In vivo	T10	Quercetin	BBB locomotion test, HE staining, immunohistochemistry, electrophysiological tests, q-RT-PCR, Western Blot
Mountney et al., 2013 [17]	USA	In vivo	T9	Sialidase and chondroitinase ABC (ChABC)	BBB locomotion test, ladder crossing test, baroreceptor regulation tests, histology, immunohistochemistry
Maqueda et al., 2019 [18]	Spain	In vivo	T8	H_2_O_2_-preconditoned human adipose mesenchymal stem cell (hAMSCs) cells (hHC016)	BBB locomotion test, CatWalk-assisted gait test, sensory tests (Von Frey and Hargreaves tests), histology, ECy myelin staining, immunohistochemistry
Rong et al., 2019 [19]	China	In vivo	T9–T11	Harpargide	BBB locomotion test, immunofluorescence, flow cytometry, Western Blot, Nissl staining, TUNEL staining
Grosso et al., 2014 [20]	USA	In vivo	T8	Liposome-encapsulated clodronate with rolipram and ChABC	Grid walk test, lesion and cavity area measurements, neuronal tracing, immunohistochemistry
Krityakiarana et al., 2016 [21]	USA	In vivo	T12	Insulin-like growth factor 1 and transferrin (TSC1)	Immunohistochemistry, cell counting
Bonilla et al., 2021 [22]	Spain	In vitro and in vivo	T8	Polyacetyl-curcumin nanoconjugate (PA-C) & human neural stem cells derived from induced pluripotent stem cells (iPSC-NSCs) & human mesenchymal stem cells (MSCs) (PA-C + iPSC-NSC + MSC)	BBB locomotion test, CatWalk gait test, immunohistochemistry, histology, Luxol fast blue staining
Karalija et al., 2014 [23]	Sweden	In vivo	C3	N-acetyl-cysteine and acetyl-L-carnitine (NAC and ALC)	Immunohistochemistry, Western Blot, Fast Blue neuronal labelling
Karalija et al., 2012 [24]	Sweden	In vivo	L5	N-acetyl-cysteine and acetyl-L-carnitine (NAC and ALC)	Immunohistochemistry, Western Blot, Fast Blue neuronal labelling
Pallier et al., 2015 [25]	UK	In vivo	T12	Fortasyn^®^ Connect (FC)	BBB locomotion test, immunohistochemistry
Zhang et al., 2016 [26]	China	In vivo	T10	Y27632 (a ROCKII inhibitor) and TDZD-8 (a GSK-3β inhibitor)	BBB locomotion test, SEP monitoring, TUNEL assay, neuronal tracing, immunohistochemistry
Wang et al., 2014 [27]	China	In vivo	T10	Etanercept (a TNF-α antagonist) administration prior NSC transplant	BBB locomotion test, electrophysiological analysis, histology, immunofluorescence, Western Blot, TUNEL staining, Nissl staining, HE staining, toluidine blue staining
Salarinia et al., 2020 [28]	Iran	In vivo	T10	Platelet-rich plasma (PRP) and mesenchymal stem cells (MSCs)	BBB locomotion test, flow cytometry, Real time-PCR, immunohistochemistry, TUNEL staining
Liu et al., 2013 [29]	Canada	In vivo	T6–T7	K_2_(QL)_6_K_2_, (QL6) a self-assembling peptide	BBB locomotion test, immunocytochemistry, electrophysiological analysis, Luxol fast blue staining, HE staining, TUNEL staining, neuronal tracing
Garcia-Álvarez et al., 2015 [30]	Spain	In vitro and in vivo	T9	IG20 a synthetic glycolipid	BBB locomotion test, mass spectrometry, RT-PCR, immunoprecipitation assays, Western Blot, immunocytochemistry
Sakka et al., 2014 [31]	Finland	In vitro and in vivo	T9–T10	NX210 oligopeptide	BBB locomotion test, open arena test, reflex testing, immunostaining
Tsai et al., 2019 [32]	Taiwan	In vivo	T9	Conditioned medium from mesenchymal stem cells (MSCcm)	BBB locomotion test, Western Blot, immunohistochemistry
Zhou et al., 2017 [33]	China	In vitro and in vivo	T9–T10	Probucol (a bisphenol compound)	BBB locomotion test, inclined plane test, HE staining, Nissl staining, Western Blot, immunofluorescence, TUNEL staining
Liang et al., 2014 [34]	China	In vivo	Th8	Neural stem cell conditioned medium (NSCM)	BBB locomotion test, neuronal tracing, Fast blue and cresyl violet staining, immunostaining
Wang et al., 2014 [35]	China	In vivo	T9	Curcumin	BMS scoring, immunohistochemistry, Western Blot, ELISA test
Sun et al., 2013 [36]	China	In vitro and in vivo	T9	Cholinergic neuron-like cells derived from BMSCs induced by D609	BMS scoring, footprint analysis, inclined plane test, swim test, Western Blot, immunohistochemistry, cresyl violet staining
Machova-Urdzikova et al., 2017 [37]	Czech Republic	In vivo	T8	Epigallocatechin gallate (EGCG)	BBB locomotion test, flat beam test, Plantar test, Rotarod test, histology, immunohistochemistry, q-RT-PCR
Sahu et al., 2018 [38]	China	In vivo	T7–T9	Ursolic acid (UA, 3-beta-hydroxyurs-12-en-28-oic acid) a mimetic of HNK-1	BMS scoring, beam walking test, Western Blot, immunohistochemistry, ELISA test
Ahmed et al., 2014 [39]	UK	In vivo	T8	Decorin	Immunohistochemistry, Western Blot, zymography
Yin et al., 2013 [40]	China	In vitro and in vivo	T9	Methylprednisolone (MP) and rolipram	BBB locomotion test, grid walking test, histology, neuronal tracing, Western Blot, ELISA tests, immunofluorescence, cresyl violet eosin staining
Yuan et al., 2012 [41]	China	In vitro and in vivo	T8	Ethyl pyruvate	BBB locomotion test, rung horizontal-ladder test, footprint analysis, immunohistochemistry, BrdU cell proliferation assay, Western Blot, TUNEL staining, neuronal tracing
Pan et al., 2014 [42]	China	In vivo	T7–T9	Tegaserod, a polysialic acid mimetic	BMS scoring, single-frame motion analysis, histology, immunohistochemistry, HE staining
Lee et al., 2012 [43]	Taiwan	In vivo	T10	Delayed granulocyte colony-stimulating factor (G-CSF)	BBB locomotion test, immunohistochemistry, histology, ELISA test, HE staining, electrophysiological tests
Aleksić et al., 2014 [44]	Serbia	In vivo	T10–T11	Thermomineral water	BMS scoring, immunohistochemistry
Wang et al., 2020 [45]	China	In vitro and in vivo	T9	Metformin	BBB locomotion test, HE staining, Nissl staining, Western Blot, immunofluorescence, TUNEL staining
Wang et al., 2017 [46]	China	In vitro and in vivo	T9–T10	Crocetin	BBB locomotion test, contact plantar placement test, immunofluorescence, ELISA tests
Zhao et al., 2016 [47]	Mongolia	In vitro and in vivo	T10	Neural stem cell transplantation and erythropoietin (NSC + EPO)	BBB locomotion test, immunohistochemistry
Li et al., 2016 [48]	China	In vitro and in vivo	T10	Flavopiridol	BBB locomotion test, flow cytometry, HE staining
Chen et al., 2021 [49]	USA	In vivo	C5	Biomaterial bridge made of poly(lactide-co-glycolide) (PLG) loaded with a lentivirus encoding IL-10	Ladder beam test, histology, immunohistochemistry, electromyogram recordings
Zhou et al., 2022 [50]	China	In vitro and in vivo	T10	FPAaF micelles (made by conjugating Fer-1 and DBCO modules to amphiphilic polymers, then click chemistry using azido linker (3-(azidomethyl)-4-methyl-2,5-furandione, AzMMMan)-modified aFGF)	BBB locomotion test, Western Blot, RNA sequencing, HE staining, immunofluorescence, histology
Sperling et al., 2019 [51]	Brazil	In vivo	T10	Galantamine	BBB locomotion test, immunohistochemistry, HE staining, flow cytometry
Ishii et al., 2012 [52]	Japan	In vivo	Th9–Th10	Adoptive transfer of type 1 helper T (Th1)-conditioned cells	BMS scoring, inclined plane test, tactile sensation test, neuronal labelling, immunohistochemistry, flow cytometry, ELISA test
Bimbova et al., 2018 [53]	Slovakia	In vivo	Th9	Atorvastatin	BBB locomotion test, ELISA test, immunohistochemistry, cell counting, RT-PCR
Huang et al., 2016 [54]	China	In vivo	T10	Tetramethylpyrazine (TMP)	BBB locomotion test, cresyl violet staining, q-RT-PCR, Western Blot, immunohistochemistry, TUNEL staining, DNA fragmentation assay, in situ hybridisation

Notes: BBB, Basso-Beattie-Bresnahan; HE, haematoxylin eosin; TUNEL, terminal deoxynucleotidyl transferase biotin-dUTP nick end labelling; q-RT-PCR, quantitative real time polymerase chain reaction; BrdU, bromodeoxyuridine; ECy, eriochrome cyanine; ChABC, chondroitinase ABC; ROCKII, Rho-Rho-associated coiled-coil containing protein kinase 2; TDZD-8, 4-benzyl-2-methyl-1,2,4-thiadiazolidine-3,5-dione; GSK3-β, glycogen synthase kinase-3β; SEP, somatosensory evoked potential; TNF-α, tumour necrosis factor-α; NSC, neural stem cell; PCR, polymerase chain reaction; RT-PCR, real time-PCR; BMS, Basso Mouse Scale; ELISA, enzyme-linked immunosorbent assay; BMSC, bone marrow stromal cells; D609, tricyclodecane-9-yl-xanthogenate; HNK-1, human natural killer-1; IL-10, interleukin-10; Fer-1, ferrostatin-1; DBCO, dibenzocyclooctyne; aFGF, acidic fibroblast growth factor; DNA, deoxyribonucleic acid. None of the studies included in this review used only in vitro methodology to examine the effects of the treatment on the improvement of SCI.

**Table 2 cells-11-03409-t002:** Summary of the BBB scores for control and treatment groups from the studies that performed the BBB test.

Author	Day	BBB Score—Control Group	BBB Score—Treatment Group
Wang et al., 2017 [16]	D7	4.5 ± 0.2 (G) n = 10	5.9 ± 0.5 (G) n = 10
Hou et al., 2021 [12]	D14	5.5 ± 0.4 (G) n = 3	11.0 ± 0.9 (G) n = 3
Wang et al., 2020 [45]	D14	2.4 ± 0.8 (G) n = 10	5 ± 0.4 (G) n = 10
Jin et al., 2021 [13]	D28	8.0 ± 0.8 (G) n = 6	11.0 ± 1.5 (G) n = 6
Caglar et al., 2018 [11]	D28	Group 7: 2.0 ± 0.1 n = 4	Group 8: 19.0 ± 0.1 n = 4
Huang et al., 2016 [54]	D28	11 ± 0.577 n = 4	15.375 ± 0.854 n = 4
Pallier et al., 2015 [25]	D28	9 ± 1.2 (G) n = 9	17 ± 1.8 (G) n = 9
Wang et al., 2014 [27]	D28	2.7 ± 0.3 (G) n = 6	ET: 5.5 ± 0.8 (G) n = 6 NSC: 7.3 ± 0.9 (G) n = 27 NSC + ET: 9.6 ± 1.2 (G) n = 27
Garcia-Álvarez et al., 2015 [30]	D28	12.4 ± 0.5 (G) n = 5	15.5 ± 1.1 (G) n = 5
Sakka et al., 2014 [31]	D28	11.8 ± 0.5 (G) n = 8	13.8 ± 0.7 (G) n = 8
Zhou et al., 2017 [33]	D28	12 ± 0.4 (G) n = 5	15.5 ± 0.9 (G) n = 5
Liang et al., 2014 [34]	D28	18.9 ± 1.0 n = 20	NSCM: 16.2 ± 0.5 n = 23 Control medium: 10.1 ±1.5 n = 22
Mountney et al., 2013 [17]	D35	12.5 ± 0.5 (G) n = 12	ChABC: 12.5 ± 0.2 (G) n = 10 Sialidase: 15.5 ± 0.6 (G) n = 13 ChABC + Sialidase: 12.5 ± 0.2 (G) n = 9
Salarinia et al., 2020 [28]	D35	10.5 ± 0.7 (G) n = 12	PRP: 12.2 ± 0.4 (G) n = 12 AD-MSCs: 12 ± 0.7 (G) n = 12 AD-MSCs + PRP: 12.3 ± 1.2 (G) n = 12
Yuan et al., 2012 [41]	D35	12 ± 0.6 (G) n = 8	0.431 mmol kg^−1^ of Ethyl pyruvate at 0 h after SCI: 17 ± 0.6 (G) n = 8
Lee et al., 2012 [43]	D37	7.1 ± 1.2 n = 5	9.8 ± 0.8 n = 5
Rong et al., 2019 [19]	D42	9.1 ± 0.8 (G) n = 6	13.5 ± 0.8 (G) n = 6
Tsai et al., 2019 [32]	D42	4.5 ± 0.5 (G) n = 3	8.5 ± 0.9 (G) n = 3
Wang et al., 2017 [46]	D42	7.0 ± 1.2 (G) n = 6	11.5 ± 1.2 (G) n = 6
Li et al., 2016 [48]	D42	9.1 ± 4.1 n = 11	13.9 ± 4.0 n = 12
Sperling et al., 2019 [51]	D42	10.21 ± 0.4 (G) n = 12	15.5 ± 0.6 (G) n = 16
Bimbova et al., 2018 [53]	D42	9.7 ± 0.9 n = 9	11.06 ± 0.9 n = 9
Liu et al., 2013 [29]	D56	6.8 ± 1.0 (G) n = 23	8 ± 1.0 (G) n = 23
Maqueda et al., 2019 [18]	D56	11.25 ± 0.25 n= 8	hAMSC: 11.38 ± 0.26 n = 8 hHC016: 15.79 ± 0.49 n = 12
Ma et al., 2020 [15]	D56	10.50 ± 0.34 n = 10	tFNA: 15.33 ± 0.31 n = 10 NSCs: 15.83 ± 0.33 n = 10 tFNA + NSCs: 18.67 ± 0.31 n = 10
Zhang et al., 2016 [26]	D56	SCI only group 8 ± 1.0 (G) n = 15	Y27632: 9.7 ± 0.4 (G) n = 15 TDZD-8: 9.7 ± 0.4 (G) n = 15 TDZD-8 + Y27632: 9.8 ± 0.3 (G) n = 15
Yin et al., 2013 [40]	D56	12.2 ± 0.2 (G) n = 8	Rolipram + MP: 15.3 ± 0.2 (G) n = 8 MP: 13.5 ± 0.2 (G) n = 8 Rolipram: 13.4 ± 0.2 (G) n = 8
Zhao et al., 2016 [47]	D56	7.94 ± 0.35 n = 10	EPO: 8.06 ± 0.31 n = 10 NSC: 10.0 ± 0.31 n = 10 NSC + EPO: 11.61 ± 0.46 n = 10
Zhou et al., 2022 [50]	D56	5.6 ± 0.3 (G) n = 3	FP + AaF: 8.8 ± 0.5 (G) n = 3 aFGF: 8.8 ±0.3 (G) n = 3 FP: 8.9 ± 0.3 (G) n = 3
Bonilla et al., 2021 [22]	D63	10.6 ± 0.8 (G) n = 16	iPSC-NSC: 10.6 ± 0.8 (G) n = 8 MSC: 11 ± 0.9 (G) n = 11 PA-C: 11.6 ± 0.9 (G) n = 6 iPSC-NSC + MSC + PA-C: 10.9 ± 1.0 (G) n = 7
Machova-Urdzikova et al., 2017 [37]	D63	6.8 ± 1.2 (G) n = 12	6.6 ± 1.0 (G) n = 12
Wang et al., 2018 [14]	D294	Con-M: 11.4 ± 0.8 (G) n = 5 PBS: 8.8 ± 0.9 (G) n = 5	13.5 ± 0.6 (G) n = 5

Notes: The treatment group with the highest BBB score achieved is reported for studies with multiple treatment groups such as the studies by Caglar et al. [11] and Yuan et al. [41]. BBB scores for studies with combined treatments are reported alongside BBB scores for each treatment individually. The timepoint for the final BBB score reported is recorded in days (D) and sample numbers for each group are also reported (n). Values estimated from graphs are indicated by (G). For the study by Pallier et al. [25], only the BBB results from Study 2 are included in this table. The BBB test was performed in 32/44 of the studies, it was not reported to be performed in the following studies: Grosso et al. [20], Krityakiarana et al. [21], Karalija et al. [23], Karalija et al. [24], Wang et al. [35], Sun et al. [36], Sahu et al. [38], Ahmed et al. [39], Pan et al. [42], Aleksić et al. [44], Chen et al. [49] and Ishii et al. [52]. ET, etanercept; NSC, neural stem cell; NSCM, neural stem cell conditioned medium; chABC, chondroitinase ABC; PRP, platelet rich plasma; AD-MSC, adipose derived mesenchymal stem cells; SCI, spinal cord injury; hAMSC, human adipose mesenchymal stem cells; hHC016, h2O2-preconditioned human adipose mesenchymal stem cells; tFNA, tetrahedral framework nucleic acid; Y27632, (a ROCKII inhibitor); TDZD-8, 4-benzyl-2-methyl-1,2,4-thiadiazolidine-3,5-dione; MP, methylprednisolone; EPO, erythropoietin; FP, Fer-1 conjugated amphiphilic polymers; AaF, AzM-aFGF; aFGF, acidic fibroblast growth factor, iPSC-NSC, human neural stem cells derived from induced pluripotent stem cells; MSC, mesenchymal stem cell; PA-C, pH-responsive polyacetal–curcumin nanoconjugate; Con-M, control medium; PBS, phosphate-buffered saline. Functional recovery greater than controls. The summary evaluation, indicated by the black diamond in the forest plot and the Z statistic, demonstrated that the overall effect of these treatments was statistically significant with a mean difference of 4.76, 95% CI [−1.85, 11.38], *p* < 0.00001.

**Table 3 cells-11-03409-t003:** Immunohistochemical outcomes in studies.

Author	Marker	Level of Expression/ Regulation	Description
Hou et al., 2021 [12]	GFAP MAP2 GAP43 MBP Iba-1 CD68 CD163	↓ ↑ ↑ ↑ ↓ ↓ ↑	GFAP decreased in treatment group suggesting treatment to inhibit reactive astrogliosis. MAP2 staining in the treatment group demonstrated a shorter distance between the neurons and the lesion centre compared to control indicating neuronal regrowth/survival. Increased GAP43-positive axons in the treatment group. GAP43 is involved with nerve regeneration. MBP increased in the lesion site of the treatment group suggesting remyelination. Iba-1 (microglia marker) decreased, CD163 (M2-associated marker) increased, CD68 (microglia activation marker) decreased in treatment group, thus treatment reduced microglia activation and promoted microglia polarization towards M2 phenotype.
Jin et al., 2021 [13]	NF-200 C-caspase-3	↑ ↓	NF-200 positive axons were increased in treatment group suggesting promotion of axon regrowth/regeneration. Decrease in C-caspase-3 suggested a decrease in apoptosis in the treatment group.
Wang et al., 2018 [14]	NF200 CD86 CD206	↑ ↓ ↑	NF-200 positive axons were increased in treatment group suggesting promotion of axon regrowth/regeneration. Decreased CD86+ cells. Increased CD206+ cells suggested an anti-inflammatory effect in the treatment group.
Ma et al., 2020 [15]	GFAP MBP NestinNeuN	↓ ↑ ↑ ↑	GFAP decreased in combined treatment group indicating that the treatment prevented NSC differentiating into astrocytes; astrocytes contribute to glial scar formation. MBP increased in combined treatment group suggesting treatment to promote NSC differentiation into oligodendrocytes. Nestin increased in combined treatment group suggesting more survival and proliferation of NSCs. NeuN increased in combined treatment group suggesting treatment promoted NSC differentiation into neurons.
Wang et al., 2017 [16]	GFAP 5-HT NF-200	↑ ↑ ↑	GFAP-positive cells increased in treatment group suggesting that the treatment promoted astrocyte activation. Increase in 5-HT positive neurons. Increase in NF200 suggested that the treatment promotes axonal regeneration.
Mountney et al., 2013 [17]	5-HT TH CGRP	- - ↑	Combined treatment had no difference in 5-HT expression from controls. 5-HT was used to identify descending axons from the brainstem. Combined treatment showed no difference in TH expression compared to controls. TH was used to identify catecholaminergic axons. Combined treatment showed an increase in expression of CGRP. This is a marker for a subtype of sensory axons.
Maqueda et al., 2019 [18]	GFAP Iba-1 Fibronectin and NF200 Reca1	↑ ↓ ↑ ↑	Treatment group had an increase in GFAP expression, however results showed that astroglial reactivity was limited compared to control group. Iba-1 expression was significantly higher in control groups compared to the treatment group suggesting a reduced microglial activity in the treatment group. Increase expression of fibronectin and NF200 in treatment group suggested an increase in the formation of matrix material and axonal sprouting. Increase in Reca1 suggested that the treatment promoted vascularisation.
Rong et al., 2019 [19]	NeuN NF200 GFAP	↑ ↑ ↓	Increased NeuN staining in treatment group suggesting an increased number of motor neurons. NF200 increased and GFAP decreased in treatment group compared to SCI injury control group suggesting treatment to promote axon regeneration and inhibit glial scar formation.
Grosso et al., 2014 [20]	ED-1	↓	Decreased ED-1 positive macrophages in the combined treatment group compared to control groups suggesting treatment to reduce macrophage accumulation and thus neuroinflammation.
Krityakiarana et al., 2016 [21]	HSP-70 HSP-32 Nestin NG2	↑ ↑ ↑ ↑	HSP-70 in grey matter (neuronal cell bodies) was increased in treatment compared to control groups. HSP-70 repairs/degrades polypeptides that have denatured due to cellular stress. HSP-32 expressing cells were increased in treatment group. HSP-32 is involved with controlling acute inflammation. Nestin expressing cells were increased in treatment group compared to control suggesting an increase in undifferentiated NSCs as Nestin is a NSC marker. NG2 marker was examined to determine effect of treatment on oligodendrocyte progenitor cells. NG2 was increased in treatment group suggesting potential remyelination.
Bonilla et al., 2021 [22]	GFAP β-III-tubulin NeuN and synaptophysin Iba-1 and Arginase-1	↓ ↑ ↑ * ↑ *	Decreased staining of GFAP positive areas suggesting a significantly reduced glial scar in combined treatment group. Increased staining of β-III-tubulin positive areas in combined treatment group suggesting nerve fibre preservation in combined treatment group. Co-localisation of NeuN and synaptophysin was increased significantly in combined treatment group indicating preservation of functional synapses. Co-localisation of Iba-1 and Arginase-1 was increased in combined treatment group suggesting treatment to promote polarisation of microglia towards an anti-inflammatory phenotype.
Karalija et al., 2014 [23]	OX42 5-HT	↓ ↑	OX-42 immunoreactivity was decreased with both treatment groups suggesting the treatments to alter the microglial response and thus the neuroinflammatory response. 5-HT positive axons were increased in both treatment groups compared to control suggesting axonal sprouting.
Karalija et al., 2012 [24]	MAP2 and synaptophysin NF GFAP OX42	↑ ↑ - ↓	Both treatment groups increased immunoreactivity for MAP2 and synaptophysin compared to control suggesting improved preservation of dendritic branches and synaptic boutons. NF was increased in both treatment groups compared to control suggesting axonal sprouting. GFAP staining showed no difference in treatment compared to controls suggesting no effect on reactive astrocytes. OX42 was decreased in both treatment groups compared to controls suggesting a reduced microglial response.
Pallier et al., 2015 [25]	NeuN APC Iba-1 GFAP	↑ ↑ ↓ ↓	Increased NeuN staining in treatment compared to controls indicating a reduced loss of NeuN-positive cells suggesting neuronal protection/survival. Increased APC staining indicating a reduced loss of APC-immunoreactive oligodendrocytes in treatment group compared to controls. Decreased staining for Iba-1 in treatment compared to controls suggesting decreased microglia activation. Decreased GFAP staining in treatment group suggesting decreased reactive astrogliosis.
Zhang et al., 2016 [26]	GAP-43	↑	Significantly increased GAP-43 staining in combined treatment one week after SCI suggesting axon regeneration.
Wang et al., 2014 [27]	GFP NF200	↑ ↑	Increased number of GFP positive NSCs in the combined treatment group indicating increased NSC survival. Increased NF200 in combined treatment group suggesting promotion of nerve regeneration.
Salarinia et al., 2020 [28]	NF200	↑	Number of NF200 positive axons in combined treatment group was significantly increased compared to controls indicating axon regeneration.
Liu et al., 2013 [29]	GFAP Iba-1 Caspase-3	↓ ↓ ↓	Decreased levels of GFAP Decreased Iba-1 compared to controls suggesting treatment reduces astrogliosis and inflammation. Reduced caspase-3 staining in treatment group suggesting treatment reduces post-traumatic apoptosis.
Garcia-Álvarez et al., 2015 [30]	GFAP NF MBP	↓ ↑ ↑	Decreased GFAP-positive cells in treatment group suggesting inhibition of astroglial cells. Presence of NF-positive cells in treatment group which were not observed in controls suggesting the presence of cortical neurons after inhibition of astroglial cells (which prevented neural precursor cell proliferation). NF staining was also increased in dorsal root ganglion cultures indicating the treatment to induce axonal outgrowth. Increased MBP in treatment group suggesting increase in oligodendrocyte MBP expression.
Sakka et al., 2014 [31]	NF	↑	Increased NF staining in treatment group suggesting neurite regrowth.
Tsai et al., 2019 [32]	β-III-tubulin	↑	Increased β-III-tubulin staining in treatment group which suggested preserved nerve fibres (axons).
Zhou et al., 2017 [33]	Nrf2	↑	Increased expression of Nrf2 in the treatment group suggested the treatment to activate the Akt/Nrf2/ARE signalling pathway therefore inducing anti-inflammatory and antioxidant responses.
Liang et al., 2014 [34]	C-caspase-3 NeuN	↓ ↑	Decreased C-caspase-3 positive neurons compared to control suggesting treatment to inhibit neuronal apoptosis. Increased NeuN-positive cells in treatment group suggesting treatment to protect/prevent neuronal loss.
Wang et al., 2014 [35]	GFAP and Nestin Iba-1 NF200	↓ ↓ ↑	Decreased GFAP and Nestin positive areas in treatment group compared to control suggesting a reduced amount of reactive astrogliosis with treatment application. Decreased amount of Iba-1 positive cells in treatment group suggesting inhibition of macrophage/microglia activation and thus inflammation in the lesion site. Increase in NF200 stained areas within the lesion site in treatment group suggesting increased number of neurons/axons and therefore neuron and axon protection.
Sun et al., 2013 [36]	NeuN and NF200	↑	Increased NeuN and NF staining in treatment group compared to controls suggesting neuron preservation and axon regeneration.
Machova-Urdzikova et al., 2017 [37]	GAP43	↑	Increased number of GAP43 positive fibres in treatment group compared to control suggesting axonal sprouting and thus axon regeneration.
Sahu et al., 2018 [38]	GFAP Iba-1 5-HT and NF200	↓ ↓ ↑	Decreased GFAP staining in treatment group suggesting reduced astrogliosis.Decreased Iba-1 immunoreactivity in treatment group suggesting treatment to attenuate microglia/macrophage activation and thus inflammation. Increased 5-HT and NF200 staining in treatment group indicating 5-HT nerve reinnervation and axonal regrowth.
Ahmed et al., 2014 [39]	GAP43 GFAP	↑ ↓	Increased number of GAP43 positive axons in treatment group suggesting axonal regrowth. Decreased GFAP staining in treatment group suggesting treatment to inhibit reactive astrocytes.
Yin et al., 2013 [40]	GFAP	↓ *	Decreased GFAP staining with combined treatment suggesting treatment to impair astrogliosis.
Yuan et al., 2012 [41]	GFAP CPSG Iba-1	↓ ↓ ↓	Decreased GFAP positive cells in peri-lesion areas compared to controls suggesting inhibition of reactive astrogliosis. Decreased size of CPSG immunoreactive area in treatment group suggesting reduction in glial scar size. Decreased Iba-1 immunoreactive cells in peri-lesion area in treatment group suggesting inhibition of microglial activation thus inflammation.
Pan et al., 2014 [42]	GFAP NeuN 5-HT and TH	↓ ↑ ↑	Decreased GFAP immunoreactivity in treatment group suggesting reduced astrogliosis. Increased NeuN positive neurons in treatment group compared to control suggesting neuronal survival. Increased 5-HT and TH axonal staining rostral to lesion in treatment group suggesting prevention of dieback or regrowth of serotonergic neurons.
Lee et al., 2012 [43]	Iba-1	↓	Decreased Iba-1 positive cells in treatment group suggesting inhibition of infiltration of microglia and macrophages.
Aleksić et al., 2014 [44]	TH NeuN GFAP Iba-1	↑ - - ↓	Increased numbers of TH positive nerve fibres in treatment group compared to control indicating catecholaminergic axon regrowth/resprouting. No difference in NeuN positive neuron numbers in treatment or control group suggesting no effect on survival of neurons. No significant difference in GFAP scar staining suggesting no influence on glial scar. Decreased Iba-1 expression in treatment group but not statistically significant compared to control suggesting potential influence on microglia/macrophage response.
Wang et al., 2020 [45]	Acetylated tubulin and Tyrosinated tubulin NeuN and GAP43 HO-1 and NQO1	↑ ↑ ↑	Increased acetylated and tyrosinated tubulin ratios in neurons in the treatment group suggesting treatment to influence microtubule stabilisation and thus induce axon regeneration. Increased NeuN and GAP43 expression suggesting treatment to promote axon regeneration.Increased HO-1 and NQO1 in treatment group suggesting treatment to increase expression of antioxidants through activating the Nrf2/ARE pathway and therefore reduce oxidative stress.
Wang et al., 2017 [46]	5-HT	↑	Increased amount of 5-HT positive nerve terminals in treatment group suggesting treatment to promote neuronal repair.
Zhao et al., 2016 [47]	NF200	↑	Increased NF200 staining in combined treatment group suggesting axon regeneration due to increased number of axons and nerve fibres.
Chen et al., 2021 [49]	NeuN NF200, GAP43 and bungarotoxin	↑ -	Increased NeuN positive neurons indicating treatment to promote neuronal and axonal regeneration/survival. No difference in NF200, GAP43 and bungarotoxin staining of reinnervation of motor end plates between uninjured control and treatment group suggesting treatment to prevent denervation of neuromuscular junctions.
Zhou et al., 2022 [50]	GFAP and NeuN	↓ ↑	Decreased GFAP Increased NeuN staining in treatment group Suggests treatment allows for neuroplasticity and neuroregeneration.
Sperling et al., 2019 [51]	NF-M and β-III-tubulin GFAP	↑ -	Increased NF-M and slightly increased β-III-tubulin expression in treatment group suggesting axon regeneration/preservation. No difference in GFAP staining between control and treatment group suggesting no difference in glial scar formation, but there was increased GFAP staining in sham group compared to treatment which indicated glial scar formation in sham group and possible reduced glial scar formation in treatment group.
Ishii et al., 2012 [52]	5-HT GFAP	↑ -	Increased 5-HT positive fibres in treatment group indicating regeneration of serotonergic nerve fibres. No difference in GFAP staining between treatment and control groups therefore no difference in glial scar formation.
Bimbova et al., 2018 [53]	ED-1 Caspase-3 GFAP Iba-1 GAP43NF	↓ ↓ ↓ ↓ ↑ ↑	Decreased ED-1 positive macrophages in treatment group at 24 h suggesting a reduced amount of infiltrating activated macrophages. Decreased Caspase-3 staining at 24 h in treatment group suggested decreased apoptosis of oligodendrocytes, astrocytes, and neurons. Decreased GFAP in treatment group compared to controls suggesting reduced astrogliosis. Decreased Iba-1 in treatment group compared to controls seen at 24 h suggesting decreased activation of microglial cells. However, no clear difference in Iba-1 staining in treatment and controls after six weeks. Increased GAP43 staining in treatment group indicating axonal growth. Increased NF staining of axons observed in treatment group suggesting regeneration.
Huang et al., 2016 [54]	PTEN and PDCD4	↓	Decreased PTEN and PDCD4 staining at day 3 in treatment group. PTEN inhibition has been shown to have regenerative effects and inhibition of PDCD4 has been shown to protect against hypoxia induced apoptosis.

Notes: ↑ indicates increased staining/expression, ↓ indicates decreased staining/expression, - indicates no difference in staining/expression, assumed results from images are indicated by the symbol *. For the study by Pallier et al. [25] only the results from Study 2 are included in this table. Two studies by Caglar et al. [11] and Li et al. [48] did not report performing immunohistochemistry, thus were not included in this table. GFAP, glial acidic fibrillary protein; MAP2, microtubule-associated protein 2; GAP43, growth-associated protein 43; MBP, myelin basic protein; Iba-1, ionized calcium adaptor molecule 1; CD68, cluster of differentiation 68 (a microglial activation marker); CD163, cluster of differentiation 163 (a macrophage M2-associated marker); NF-200, neurofilament 200; C-caspase-3, cleaved caspase-3; CD86, cluster of differentiation 86 (an activated macrophage M1 marker); CD206, cluster of differentiation 206 (an activated macrophage M2 marker); Nestin, a marker of progenitor cells/stem cells; NeuN, neuronal nuclear protein (marker for neurons); NSC, neural stem cell; 5-HT, 5-hydroxytryptamine (serotonin); TH, tyrosine hydroxylase; CGRP, calcitonin gene related peptide; Reca1, marker for endothelial cells; SCI, spinal cord injury; ED-1, marker for macrophages (CD68); HSP-70, heat shock protein 70; HSP-32, heat shock protein 32; NG2, neuron-glial antigen-2; OX42, microglial marker; NF, neurofilament; APC, adenomatous polyposis coli tumour suppressor protein; GFP, green fluorescent protein; Nrf2, nuclear erythroid 2-related factor 2; CSPG, chondroitin sulphate proteoglycans; HO-1, heme oxygenase-1; NQO1, NADH dehydrogenase quinone 1; NF-M, neurofilament M; PTEN, Phosphatase and TENsin homolog; PDCD4, programmed cell death protein.and therefore an increase in staining suggests the treatment to promote axonal regeneration and/or survival of neurons [49].

## Data Availability

All data generated as part of this study are included in the article.

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
