# Peer review of "Experimental Treatments for Spinal Cord Injury: A Systematic Review and Meta-Analysis"

_cells, 2022, doi:10.3390/cells11213409_

Round 1

Reviewer 1 Report (Previous Reviewer 1)

As mentioned in my first report, this is a valuable contribution to the field. The revisions are useful additions. 

Reviewer 2 Report (Previous Reviewer 2)

we thank the reviewers for their resubmission, still the write up of this current reads like a scoping review, In literature, there are couple of recent meta analysis that are discussing the same topic./.

Novelty is a major concern,

This manuscript is a resubmission of an earlier submission. The following is a list of the peer review reports and author responses from that submission.

Round 1

Reviewer 1 Report

The authors provide a very useful analysis of a selection of SCI treatments with promising translational potential. The tables and figures are give the reader both a quick idea of treatment variables and outcome results, as well as material for more in-depth examination, supported by the main text. The relatively small number of studies that was possible to include, and their variability in terms of details limit the impact of the paper, but it, nevertheless, is highly relevant for basic and clinical scientists in the field.

Comment

One would like that the authors suggest some more specific propositions for future research, which might contribute to validate the examined treatments options or other/novel therapeutic strategies. 

Reviewer 2 Report

the authors aimed to systematically review studies published in the area of experimental treatments with neuroregenerative and neuroprotective abilities to improve spinal cord injury conditions.

the topic is of interest to clinicians and to researchers however the way that this systematic review is conducted harbors a number of flaws.

major comments

the authors have selected 3 databases (Pub Med, Web of Science, and Ovid Embase)  with no justification for not including other medical-related databases such as the EMBASE database coupled with a deficient ad limited search strategy mesh words that may have missed several studies.

For example, the authors have used spinal cord injury) AND (treatments) AND (neuroprotection) AND (axon  regeneration) ignoring other mesh terms such as rehabilitation, neural regeneration, neurogenesis, SCI.. etc

Second in the inclusion-exclusion criteria, one notices that in vtro and in vivo anmal studies were included which by itself a bisa towards analysing the outcomes.

Third, the exclusion criteria excluded non-experimental studies: what is mennt by this? do they mean clinical studies?

also, they included this vague sentence: "only studies reporting a novel experimental treatment for SCI were included, studies attempting to en hance delivery of a known treatment or replicate results from a previous study were not included" how could they discern between novel or non-novel?

the whole article reads as if it is a review article and not a meta-analysis, this needs to be changed into a review or a scoping review